# Application of Biological Domain Knowledge Based Feature Selection on Gene Expression Data

**DOI:** 10.3390/e23010002

**Published:** 2020-12-22

**Authors:** Malik Yousef, Abhishek Kumar, Burcu Bakir-Gungor

**Affiliations:** 1Department of Information Systems, Zefat Academic College, Zefat 13206, Israel; 2Galilee Digital Health Research Center (GDH), Zefat Academic College, Zefat 13206, Israel; 3Institute of Bioinformatics, International Technology Park, Bangalore 560066, India; abhishek@ibioinformatics.org; 4Manipal Academy of Higher Education (MAHE), Manipal 576104, India; 5Department of Computer Engineering, Faculty of Engineering, Abdullah Gul University, Kayseri 38080, Turkey; burcu.gungor@agu.edu.tr

**Keywords:** feature selection, feature ranking, grouping, clustering, biological knowledge

## Abstract

In the last two decades, there have been massive advancements in high throughput technologies, which resulted in the exponential growth of public repositories of gene expression datasets for various phenotypes. It is possible to unravel biomarkers by comparing the gene expression levels under different conditions, such as disease vs. control, treated vs. not treated, drug A vs. drug B, etc. This problem refers to a well-studied problem in the machine learning domain, i.e., the feature selection problem. In biological data analysis, most of the computational feature selection methodologies were taken from other fields, without considering the nature of the biological data. Thus, integrative approaches that utilize the biological knowledge while performing feature selection are necessary for this kind of data. The main idea behind the integrative gene selection process is to generate a ranked list of genes considering both the statistical metrics that are applied to the gene expression data, and the biological background information which is provided as external datasets. One of the main goals of this review is to explore the existing methods that integrate different types of information in order to improve the identification of the biomolecular signatures of diseases and the discovery of new potential targets for treatment. These integrative approaches are expected to aid the prediction, diagnosis, and treatment of diseases, as well as to enlighten us on disease state dynamics, mechanisms of their onset and progression. The integration of various types of biological information will necessitate the development of novel techniques for integration and data analysis. Another aim of this review is to boost the bioinformatics community to develop new approaches for searching and determining significant groups/clusters of features based on one or more biological grouping functions.

## 1. Introduction

Biological systems are massively complex and heterologous in nature. To resolve the mysteries behind complex biological systems, large-scale studies have been conducted which yielded massive volumes of biological data, including the genetic variations associated with specific phenotypes. Currently, we are encountering an -omics revolution in which genome, epigenome, transcriptome, and other -omics can be readily characterized. With advancements in various -omics approaches, it is now possible to generate multi-omics data to answer various biological problems. Nowadays, several types of -omics data are considered as depicted in Figure 1, and the numbers of different -omics data types are increasing day-by-day [1]. Additionally, there are complex cascades and interactions among different -omics data types. For example, genomic and epigenomic variations have the capacity to control or modulate the transcriptome and in turn affect the proteome. Here, epigenomics refers to the measurement of DNA methylation, histone modifications (methylation, acetylation, phosphorylation, DP-ribosylation, and ubiquitination), and noncoding RNAs (microRNAs, long noncoding RNAs, small interfering RNAs). Similarly, the epigenome of an organism refers to the entire collection of the molecules that modify the genome and control the genes to turn on and off. Since the epigenome shows how environmental factors influence the activity of genes, the study of the epigenome integrated with the study of the genome is crucial to fully account for phenomics. Accounting for such molecular deviations is crucial for making tangible improvements in biomarker analysis.

Traditional analyses attempted to untangle the molecular mechanisms of complex diseases using a single -omics dataset which contributes towards the identification of disease-specific mutations and epigenetic alterations. However, in the postgenomic era, it has been noticed that a single -omics dataset is not sufficient to explain disease hallmarks. It requires the combined analysis of various -omics datasets. As such, recent studies are shifting towards multi-omics data analysis, where each of these different -omics data types are critical for deciphering the molecular signatures of human diseases. Therefore, the integrated analysis of different data types has become a recent trend. For a holistic understanding of complex biological problems, it is becoming clear that integrations of different -omics data types are essential steps. However, it is a notorious task, as handling heterogeneous and noisy biological data is a challenging issue [2].

In addition to the ‘omics’ realm, another major reason for phenotypic differentiation is post-translational modifications (PTM). They can be both in physiologically reasonable and pathologically anomalous forms. Methods for bioinformatically incorporating PTM effects are emerging from the gradual improvement of sequence motifs, or less directly from compensatory expression patterns that emerge when an organism seeks to correct for aberrant biochemistry arising from anomalous structural modifications.

Due to the recent advancements in next-generation sequencing and microarray technologies, the cost of obtaining the gene expression profile of a sample is rapidly decreasing, and hence expression profiling has become a routine protocol in biological laboratories. The high turnaround of expression data is also coupled by the massive increase in the use of the revolutionary RNA-Seq method [3]. It is best exemplified by the large oncogenomic expression profiles hosted at The Cancer Genome Atlas (TCGA) [4]. Mutations are the core causative agents of diseases such as different cancers [5] when coupled with gene expression profiles. These datasets provide sufficient information to scientists and physicians for deciphering the disease mechanisms. It is becoming clear that the proper design of the RNA-seq can be used for mutational profiling as well as expression profiling [6]. This information also enables the design of platforms to assist diagnosis, to assess patients’ prognosis, and to create patient treatment plans. For instance, van’t Veer et al. had collected gene expression profiling datasets of primary breast tumors derived from a cohort of 117 young patients [7]. Machine Learning (ML) with feature selection was used to unravel a gene expression signature, which served as a signal for distant metastases, even divergent conditions such as lymph node negative [7].

Data analysis approaches to gene expression profiling have evolved rapidly as there are massive shifts from DNA microarray to RNA-seq-based profiling. The earlier methods involved clustering approaches and traditional ML approaches. Since a large volume of biological knowledge has become available, in the literature there are obvious shifts from the pure data-oriented approaches to biological domain knowledge-based integrative approaches. This fact has triggered bioinformatics researchers to suggest and develop advanced tools that consider the emerging biological knowledge, and hence they exploit this knowledge for deep analysis of the data. There are many resources of biological knowledge, such as textual knowledge, as more and more literature emerges, different databases and repositories such as miRTarBase [8] for microRNA, DNA Sequence Databases, Immunological Databases, Gene Expression Omnibus (GEO), Proteomics Resources, Protein Sequence Databases, TCGA, Gene Ontology (GO) and others.

Most feature selection algorithms that are applied on gene expression data are based on statistics and ML. However, most of them neglect the biological knowledge of the data that could contribute to perform better feature selection. R. Bellazzi and B. Zupan [9] discussed recent developments in gene expression-based analysis methods, focusing on studies (such as associations and classification) and implications (such as reverse-engineering of gene–gene networks and resulting phenotypes). Authors surveyed the clustering approaches that group the genes using different distance measures, such as Euclidean distance and/or Pearson’s correlation. Moreover, incorporating biological knowledge in the clustering algorithm is a very challenging task. The GOstats package [10] allows one to define semantic similarity between the genes via incorporating the GO [11]. An additional study by Kustra and Zagdanski [12] used the incorporation of GO annotation to expression data by inducing a correlation-based dissimilarity matrix to derive a GO-based dissimilarity matrix.

The flood of -omics data and the need for more informative results urge the need for integrative approaches. The book of Ref. [13] is the first book on integrative data analysis and visualization in this area. It outlines essential techniques for the integration of data derived from multiple sources. It is one of the first systematic books that overviews the issue of biological data integration using analytical approaches. The book provides a framework for the creation and implementation of integrative analytical methods for the study of biological data on a systematic scale. Additionally, a recent review [2] describes the principles of biological data integration along with different approaches and methods indicating the importance of utilizing ML for biomedical datasets. However, to the best of our knowledge, in the literature, there is no comprehensive survey on biological domain knowledge-based feature selection methods, except from the study of Perscheid et al. [14] that compares the performances of traditional gene selection methods against integrative ones. Moreover, authors also proposed a straightforward method to integrate external biological knowledge with traditional gene selection approaches. They introduced a framework for the automatic integration of external knowledge for selected genes and their evaluation. Herein, we aimed to provide a selective review on such gene selection approaches. In this regard, for the analysis of gene expression datasets, we present the traditional and integrative gene selection approaches in Section 2.1 and Section 2.2, respectively.

On the other hand, multi-dimensional biological data are another challenge as these data are often derived from limited numbers of samples because of the associated costs of biological data generation. This is an example of the ‘curse of dimensionality’ problem, as initially reported by Bellman [15] in 1961. It means that the dimensions of gene features and/or functional parameters are critical input variables, and there are requirements of a minimal number of samples for the estimation of an arbitrary function, where an increase in sample size improves the chance of function prediction. In Section 3 of this manuscript, we present a prospective solution to this problem by defining a novel grouping of features and estimating their contribution to the machine learning model for two-class classification problems. We evaluate the methods that select the features using a classifier in a traditional way in Section 3.1, and we present integrative approaches that incorporate biological domain knowledge into ML to group and rank the genes in Section 3.2. In Section 4, we conclude our review with discussions and future prospects.

## 2. Gene Selection Approaches for Gene Expression Datasets

Gene selection approaches for gene expression datasets can be mainly categorized into two classes, such as traditional gene selection and integrative gene selection. While traditional gene selection approaches are solely based on statistical and computational analyses of the expression levels, integrative gene selection approaches incorporate domain knowledge from external biological resources during gene selection.

### 2.1. Traditional Gene Selection

Traditional gene selection approaches are heavily based on statistical and computational analyses of the actual expression levels. Recent reviews have summarized various methods for describing the selection process of disease-specific features from large gene expression datasets [16,17]. Primarily, these approaches are classified into three major classes, as (i) filtering-based, (ii) wrapping-based, and (iii) embedding-based approaches. Briefly, the filtering approaches are based on F-statistic (ANOVA, *t*-test, etc.), not based on ML. Wrapping-based approaches are primarily learning techniques and these are used for the exploration of usefulness of features, whereas embedding-based approaches are combining the feature selection and the classifier construction. Wei Pan carried out a comparative study on different filtering methods in Ref. [17] and he summarized similar and dissimilar points among three main methods (namely *t*-test method, regression modeling approach and mixture model approach).

Additional comparisons of filtering techniques are available in Ref. [16]. I. Inza [17] also carried out a comparison between filter metrics and the wrapper sequential search procedure, which are both applied on gene expression datasets. Additionally, hybrid, and ensemble approaches, which combine multiple approaches, are two additional categories of gene selection. Cindy et al. [14] presents an overview of the recent gene selection methods, where each method is classified according to these five categories.

The traditional gene selection approach has several drawbacks. For example, the filtering approach evaluates the significance of each gene individually without considering the relationships and the interactions between the genes. Although the wrapping-based approaches can find the optimal set, it might be specific to the model used, such as SVM, decision trees or other models. In other words, it might be overfitting the data [18]. The main disadvantages of such methods are their difficulties for biological interpretation, and they are unlikely to generate new biological knowledge.

### 2.2. Integrative Gene Selection

Although the traditional gene selection approaches became popular for a long time, they have several drawbacks when one needs to precisely identify the underlying biological processes. Alternatively, integrative gene selection approaches incorporate domain knowledge from external biological resources during gene selection [9,18], which improves interpretability and predictive performance. One of the widely used external ontology resources is the Gene Ontology (GO) [19], which provides (i) cellular component (CC), (ii) molecular function (MF), and (iii) biological process (BP) terms for the products of each gene. GO captures biological knowledge in a computable form that consists of a set of concepts and their relationships to each other. The first attempt to integrate biological background into a statistical analysis/ML analyses was to incorporate Gene Ontology (GO) [19] in clustering gene expression data [10]. Another widely used external ontology resource is the Kyoto Encyclopedia of Genes and Genomes (KEGG), which is a pathway knowledge-base providing manually curated pathways [20]. Yet another widely used external biological resource is DisGeNET, which is a meta knowledge-base on gene–disease–variant associations [20].

One example of the integrative gene selection approach is proposed by Qi and Tang, where they utilize the power of biological information contained in GO annotations to rank the genes [21]. The algorithm is designed in an iterative manner that starts by applying Information Gain (IG) to compute discriminative scores for each gene. The genes that have a score of zero are removed from the analysis. The second step is to integrate the biological knowledge, which is achieved by annotating those surviving genes with a GO term. The third step is to score the GO terms as the mean of their associated genes’ discriminative scores, which were computed before using IG. The final gene set is created as follows: Starting from the highest ranked GO terms, the genes with the highest discriminative scores are chosen. These genes are removed from the annotated genes and this procedure is repeated until the final gene set is complete. Using multiple cancer datasets, Qi and Tang showed that their proposed method can achieve better results, as compared to using IG only.

Another example of the integrative gene selection approach is SoFoCles [22], which uses GO terms to find semantically similar genes. In order to assign a discriminative score to each gene, SoFoCles utilizes a classic filter approach, such as χ2, ReliefF, or IG. The initial set of candidate genes is composed of the top n ranked genes. Genes receive a similarity score based on their associated GO terms. Then, the genes which have a high similarity score, i.e., the genes that are semantically very similar to the candidate genes, are added to the set of candidates. The experiments conducted on SoFoCles showed that the incorporation of biological knowledge into the gene selection process improves the results.

Yet another study by Fang et al. [18] combines KEGG and GO terms with IG. The authors initially apply IG on the dataset as the filtering step and then check the GO and KEGG annotations of the remaining genes. Then, the authors use association mining and calculate the interestingness of the frequent itemsets by averaging the original discriminative scores (from IG) of the included genes. The final gene set is generated via selecting the highest ranked genes from the top n frequent itemsets. They evaluated this method using GO, using KEGG, and using both terms against IG only and against Qi and Tang’s approach. Although their proposed approach slightly increased the overall accuracy, the main advantage of this approach was that it used a much lower number of genes.

The integrative gene selection approach that is proposed by Raghu et al. [23] makes use of KEGG, DisGeNET, and further genetic meta information [20]. In their approach, for each gene, (i) the importance score and (ii) the gene distance metrics are computed. The importance score is calculated via combining a gene–disease association score from DisGeNET with the gene expression levels in the data. The gene distance is defined as the physical distance between two genes (in terms of their chromosomal locations) and their associations to the same diseases. Both of the scores (importance score and gene distance) are later used to find maximally relevant and diverse gene sets. As compared to variance-based gene selection techniques, the use of the top n genes according to the importance score resulted in a slightly better performance in predictive modeling task.

The integrative approach of Quanz et al. aims to map genes into KEGG pathways and then uses these pathways as features for further pattern mining [24]. In their approach, they make use of a global test to extract KEGG pathways which are related to the phenotypes of a dataset. In their feature extraction step, the genes in each pathway are then transformed into one single feature by applying mean normalization or logistic regression. In this way, the data are represented as the number of pathways, which can be considered as a feature reduction step and it provides dramatic reduction. For instance, for the diabetes data, 17 pathways, out of approximately 300 pathways, are selected and thus for the classification task the dimensionality is reduced from 22,283 to 17. Even though this approach was not tested on multiclass problems such as cancer (sub-) type classification, the experiments on binary classification problems showed an improved performance over different traditional approaches.

Mitra et al. adopted the clustering large applications based upon randomized search (CLARANS) method to the feature (gene) selection problem via utilizing biological knowledge [25]. Their reduced feature set is composed of gene clusters, which are the medoids of biologically enriched sets. Later on, the authors attempted to use a fuzzy clustering technique instead of CLARANS, and developed a technique called FCLARANS for feature selection [26].

In Ref. [27], the authors proposed an integrative gene (feature) selection approach based on the sample clustering technique, which utilizes gene annotation information from GO. On the generated gene–GO term matrix, they applied Partitioning Around Medoids clustering. In their method, the optimal number of clusters (k) is chosen by comparing their silhouette index values. For the selected k number of clusters, the medoids are used as the selected gene subset. They reported that the integration of biological knowledge during the gene selection process not only reduces the dimensionality of the feature space, but also increases the accuracy of sample classification.

The related studies that are presented until this point are highly specific to a single knowledge-base, e.g., KEGG pathway or GO terms. On the other hand, Perscheid et al. [14] proposed an approach that can flexibly combine traditional gene selection approaches with several knowledge-bases. They comparatively evaluated the performance of traditional gene selection approaches with integrative gene selection approaches. Their study concluded that the integration of external data especially improves on simple traditional filter approaches, e.g., information gain. Once external biological data are integrated, such traditional filter approaches become compatible with more complex machine learning approaches at very similar classification accuracies, but far lower computational running times and a more transparent and thus interpretable computation processes.

The above-mentioned studies proposed predictive models, but most of the time, instead of obtaining high predictive accuracies in these models, the scientists are curious about the biological meaning of the predictive model. The ‘black box’ nature of the predictive model can hamper its interpretation. The information excerpted from the model may require further processing, and careful interpretation with corresponding biological knowledge may be needed. The interpretation of the complicated cases may be quite challenging, and such an interpretation may currently be out of reach. Although the joint analysis of multiple biological data types has the potential to enlighten our understanding of complex biological phenomena, the data integration is challenging due to the heterogeneity of different data types. For example, an expression profile, as obtained from a transcriptomic study, is a vector of real values and the length of a vector is equal to the number of genes in the genome. However, the genetic variants as obtained from a genomic study are categorical, and they have different vector lengths. While different studies [1,4] proposed several strategies for data integration, the best practices by which -omics data types can be integrated and information on how to integrate these biological data are still needed.

Feature selection and discovering the molecular explanation of diseases describe the same process, where the first one is a computer science term and the second one is used in the biomedical sciences. In 2007, Yousef et al. proposed a new feature selection method, support vector machines–recursive cluster elimination (SVM-RCE), to group/cluster genes for gene expression data analysis. This study invented the “recursive cluster elimination‘‘ phrase for the first time in the machine-learning domain and introduced it to the computational community. As such, this study became a pioneer study in this field. Interests in this approach have increased over time and several studies have successfully applied the SVM-RCE approach to identify the features/genes that are directly associated with a disease/condition [28]. This growing interest is based on the reconsideration of how feature selection in biological datasets can benefit from incorporating the biomedical relationships of the features in the selection process. The usefulness of SVM-RCE then led to the development of maTE [29], which uses the same approach based on the interactions of microRNAs (miRNA) and their gene targets. Additionally, in the literature, the biological information buried in genetic interaction networks is utilized for classification studies. For example, SVM-RNE (SVM with recursive network elimination) integrates network information with recursive feature elimination based on SVM [30]. It is shown that SVM-RNE has a good performance and also improves the biological interpretability of the results. Studies similar to SVM-RCE and SVM-RNE were later carried out by different groups [31,32], which indicates the importance and the merit of the SVM-RCE approach. The study of Ref. [33] has a slightly modified SVM-RCE algorithm in the disease state prediction step. Additionally, they used the already invented term of “recursive cluster elimination”.

The study of Zhao, X. et al. [34] has used the SVM-RCE tool for comparison and used expression profiles for identifying microRNAs related to venous metastasis in hepatocellular carcinoma. Another similar study to SVM-RNE is carried out by Johannes M. et al. [35] for integration of pathway knowledge into a reweighted recursive feature elimination approach for the risk stratification of cancer patients. A recent tool, SVM-RCE-R [36], is an updated version of SVM-RCE, which is implemented in Knime [37], and uses a random forest classifier with additional important features such as suggesting a new approach of ranking the clusters.

The term “knowledge-driven variable selection (KDVS)” is a similar term to “integration of biological knowledge”, and both of them are used in the process of feature selection. An additional similar study that applied KDVS to SVM-RNE is presented by Ref. [38], in which the authors proposed a framework that uses a priori biological knowledge in high-throughput data analysis.

The RCE algorithm [28] considers similar features/genes and applies a rank function to the feature group. Since it uses k-means as the clustering algorithm, we refer to these groups as clusters, but it could include other biological or more general functions combined with the features, as was suggested in several studies [29,30]. In the original paper of SVM-RCE, the contribution to the accuracy is achieved in distinguishing specific classes for ranking the clusters. The data for that ranking are divided into training and testing, with the data represented by each gene/feature being assigned to a specific cluster of features. The rank function is then applied as the mean of m times repeats of the training–testing performance while recording different measurements of accuracy (sensitivity, specificity, etc.).

In Table 1, we summarize the specifications, advantages and disadvantages of the presented integrative gene selection approaches.

## 3. Grouping and Ranking of the Genes for Classification Problem

The genes that are involved in the same biological process are likely to be co-expressed [41]. Therefore, one potential way of discovering gene function is to group genes with a similar expression profile. Thus, different clustering algorithms [42] were considered to perform the grouping step. This was the first approach, and more advanced approaches that use biological information in order to group the genes are later proposed. In this section, we will introduce a generic approach to grouping that is accompanied by ranking and classification. The presented model is used by different studies and other similar studies are still ongoing.

The main aim of the generic approach is to search for and determine significant groups/clusters of features based on one or more biological grouping function (will be referred as *bioF()* throughout the rest of this paper) that are integrated with the ML algorithms. The generic approach is presented in Figure 2. The advantage of those systems is that the grouping of the genes/features is in the hand of the researcher, that is, it is actually based on available biological knowledge. The researcher will provide how genes or features should be grouped and then the algorithm will proceed to score and rank those groups in terms of the classification problem. The final model will be built from the top n groups according to the researcher’s settings. The outcome of the algorithm is different from the traditional current approaches (such as SVM-RFE [43]), where the algorithm takes as input the data of gene expression with class labels. Then the outcome is just a list of significant genes that are able to distinguish the two classes. With the integration framework, the researcher will get a more informative list of significant groups/clusters with its genes list that is able to distinguish the two classes. Additionally, the researcher can use the computational approach of grouping that is based on clustering approaches such as k-means or others, and specify different measurements for ranking the groups/clusters based on their interest and their research aims. The outcome of the algorithm will be more specific to the researcher’s interest.

The generic approach mainly consists of two main components. The first component is the grouping step relying on the *bioF()* function that is based on biological knowledge to group the genes into groups. For example, *bioF()* might be disease-related genes; then the function will group the genes into groups where each group is associated with one disease. Another possibility is grouping the genes that are targeted by specific miRNAs, such as in the maTE [29] tool. One interesting use of *bioF()* is that it allows one to create different biological groupings, such as creating groups related to miRNA, groups related to disease, groups related to KEGG pathways, and others. However, the grouping can also be based on clustering algorithms such as k-means, as suggested in SVM-RCE [28] for grouping correlated genes. Similarly, SVM-RNE [30] incorporates another tool, GXNA [44], to create the groups. GXNA utilizes gene expression profiles and prior biological information to suggest differentially expressed pathways or gene networks.

### 3.1. Traditional Approach of Feature/Gene Selection Using a Classifier

There are many classifiers that were used to fit the data in order to rank the features and perform the process of features selection. The most simple one is the linear model, where the coefficients of the variable/features can serve as a measurement of the feature’s rank. We will be describing the first approach that suggests the RFE procedure. RFE refers to recursive feature elimination. The approach uses SVM as the linear model.

#### SVM-RFE (Support Vector Machines with Recursive Feature Elimination)

Let us assume that we are given a set of points called S. S consists of m points xi ∈ Rd in dimension d. Let us assume that we have two class labels, denoted by yi ∈ { −1, +1}. We call S a linear separable set if there is a hyperplane of equation w·x + w0 = 0 (we refer to it as hyperplane (w, w0)) that separates the points with label + 1 from the points with label −1. The signed distance di of a point xi to the separating hyperplane (w, w0) is given by di = (w·xi + w0)/||w||.

For simplicity, let us define the optimal separating hyperplane to be f(x) = w0 + w1 × 1 + w2 × 2 + … + wd × d, where the (x1, x2, …, xd) is the features and w = (w1, w2, …, wd) is the corresponding weights. SVM is actually the solution of finding the optimal linear function, as developed by Vapnik [45]. It is obvious that the contribution of features with lower weights is non-significant to the sign of the f(x). So one can consider removing those features in order to perform dimension reduction of the feature space.

SVM-RFE, which stands for support vector machines with recursive feature elimination, was firstly introduced by Isabelle Guyon et al. [43] and applied to gene expression data. The primary goal of SVM-RFE is to use SVM (linear SVM) to compute the weights of the features. The weights are actually the ranks of the features. SVM-RFE performs an iterative step to remove features with low ranks. The RFE procedure can be described as:Train the classifier on the given data;Assign rank for each feature as its weight;Remove one feature or percentage (10%) with the smallest weight;Repeat steps 1–3 until reaching a predefined number of genes.

Different studies [46] have emerged as extended variations of the original SVM-RFE algorithm. However, SVM-RFE has some limitations that other studies have reported to suggest an improvement approach. One such limitation is that SVM-RFE is designed as a greedy method and tries to find out superlative possible combinations leading to binary classification, where these combinations may not be biologically significant. To overcome this limitation, a novel feature selection algorithm, sigFeature [47], based on SVM and t statistic, was developed.

### 3.2. Biological Domain Knowledge Based ML Approaches

ML is becoming a very powerful computational approach in the field of bioinformatics. In this respect, the first step in utilizing ML is to apply unsupervised and supervised algorithms on biological data. However, in the era of big biological data, we come up with emerging approaches that integrate biological knowledge with ML. A recent review [48] on ML and complex biological data discusses the challenges and hurdles of the analysis and discovery of complex biological data. They predict that in the very near future, more researchers will be interested in applying ML to complex biological data. In this section, we will review different approaches that consider the biological structure or knowledge for the process of feature selection. The following approaches follow the generic approach presented in Figure 2.

#### 3.2.1. SVM-RCE (Support Vector Machines with Recursive Cluster Elimination)

SVM-RCE [28,36] is based on the concept of grouping and ranking, where the k-means clustering algorithm is used for performing the related grouping of *bioF()*. Correlated genes are hypothesized to have similar biological functions. Then, the rank component is applied to assign a score for each cluster, indicating its significance in terms of the classification of the two given classes. In order to perform the rank component, each cluster is considered by representing the data based on the genes that belong to it while keeping the class labels of the original data. Now the data are transferred to cluster genes representation. The rank component performs internal cross-validation and aggregates the performance outcome as the score of the cluster. The RCE procedure is applied to remove the lowest ranked groups. The SVM-RCE results show that the classification accuracy is superior to other approaches, suggesting that the classification results are more interpretable, and this creates new hypotheses for future investigation.

#### 3.2.2. SVM-RNE (Support Vector Machines with Recursive Network Elimination)

SVM-RNE [30] is an extended version of the SVM-RCE approach that uses the tool GXNA [42] as the *bioF()* for grouping the genes into subnetworks of genes. Then, a similar procedure of ranking is applied as described in SVM-RCE. SVM-RNE also performs the recursive elimination procedure by ranking firstly all the groups, and then removing the least significant groups. The algorithm proceeds by applying again the GXNA to suggest groups. This process is repeated until satisfying some predefined constraints on the number of groups.

#### 3.2.3. MaTE

Disease development mechanisms mainly involve changes in the transcript levels and protein abundance. MicroRNAs (miRNAs) are instrumental in regulating the gene expression, and hence they affect transcript levels and protein abundance. The fact that microRNAs target more than one mRNA helps us to group the genes into groups where each group consists of the list of genes targeted by a specific microRNA. In other words, the *bioF()* biological grouping function here is the biological association between microRNA and its set of targets. The *bioF()* grouping function is based on the database mirTarBase [8]. mirTarBase has accumulated more than three hundred and sixty thousand miRNA-target interactions. Thus, a novel approach called maTE [29] has been developed.

Table 2 presents partially the result of applying *bioF()* on mirTarBase. Additionally, it performs a computation procedure in order to score/rank the importance for each group for the classification tasks.

The inputs to the maTE tool are the gene expression data, and the list of microRNAs and its target genes. The main function of the tool is to produce a group of genes based on the miRNA target information and then rank each group by applying random forest with cross-validation, which is repeated r times. The average of the accuracy for each iteration is actually the rank for a specific group. Then the groups are ranked according to the rank values. The model will be built considering the genes on the top *j* groups. The default value of *j* is 2. We apply to the training part of the data a *t*-test statistics in order to remove noisy genes. The test part of the data is used in order to estimate the performance of the tool.

#### 3.2.4. CogNet

The CogNet tool is a classification of gene expression data based on ranked active-subnetwork-oriented KEGG pathway enrichment analysis [39]. CogNet is based on biological knowledge as a function for grouping the genes for the task of ranks and classification. The pathfindR tool serves to be the biological grouping function allowing the main algorithm to rank active-subnetwork-oriented KEGG pathway enrichment analysis [49]. CogNet was tested on 13 gene expression datasets of different diseases. In these experiments, CogNet was shown to outperform maTE and obtain similar performance results with SVM-RCE.

CogNet provides a list of significant KEGG pathways, including its genes that are able to separate the classes of the data. The list would serve the biology researcher for deep analysis and better interpretability of the role of KEGG pathways in the data, or the case that is being studied. As a future work, we would develop CogNet to explore the effectiveness of different combinations of the KEGG pathways in the data. In the current version, we treat each KEGG pathway individually.

#### 3.2.5. MiRcorrNet

Due to the advances in technology, both mRNA and microRNA expression profiles can be generated allowing integrative analysis aiming to uncover the functional effects of RNA expression in complex diseases, such as cancer. Most of the approaches that integrate miRNA and mRNA are based on statistical methods, such as Pearson correlation, combined with enrichment analysis approaches. In this study [40], a novel tool is used called miRcorrNet, which performs machine learning-based integration to analyze miRNA and mRNA gene expression profiles. miRcorrNet groups mRNA genes based on their correlation to miRNA expression. Then, these groups are subjected to a rank function for classification. We have tested our tool on TCGA data miRNA-seq and mRNA-seq expression compared to other tools. The performance results show that the tool works as well as other tools in terms of accuracy measurements, reaching an AUC above 95%. Moreover, we conducted a deep biological analysis to explore the list of significant miRNAs. Accumulated results suggest that miRcorrNet is able to accurately prioritize pan-cancer-regulating high-confidence miRNAs.

## 4. Conclusions

As we have more advanced high-throughput technologies, big transcriptomic datasets become available, and extracting insights from long lists of differentially expressed genes becomes a challenge. Since the gene expression data typically have small samples size but high dimensions and noise, the major challenge is the detection of disease-related information from vast amounts of redundant data and noise. As such, the gene (feature) selection and the removal of redundant/irrelevant genes has been a key step to address this problem. For gene expression data analysis, most of the existing feature selection methods rely on expression values alone to select the genes, and biological knowledge is integrated at the end of the analysis in order to gain biological insights or to support the initial findings. However, lately, the gene selection process has shifted from being purely data-centric to more incorporative analysis with additional biological knowledge. Integrative gene selection approaches incorporate domain knowledge from external biological resources during gene selection [9,18], which improves interpretability and predictive performance. One of the more widely used external ontology resources is GO [19], which captures biological knowledge in a computable form that consists of a set of concepts and their relationships to each other. As another alternative, pathway-based analysis approaches aim to investigate the aggregation of the genes that are part of a functional unit, where these functional units are predefined by prior biological knowledge. These pathway-based methods rely on statistical tests that aim to detect damaged functionalities, which may result in disease phenotype. Several studies reported that the genetic variations occurring at multiple loci often disturb signal transduction, and regulatory and metabolic pathways, which causes severe changes in phenotype [18]. In this regard, a widely used external ontology resource is KEGG, which is a knowledge-base of manually curated pathways [20]. Yet another widely used external biological resource is DisGeNET, which is a meta knowledge-base for gene–disease–variant associations [20].

High-throughput profiling technologies currently enable us to concurrently measure gene expression levels for tens of thousands of genes in a single experiment, but they have some drawbacks. The high dimensionality of the gene expression data and relatively small sample sizes make the interpretation of the data a complicated, and often overwhelming, task. Although sample sizes have continued to grow in recent years, new and efficient feature selection algorithms are still needed to overcome the challenges in the existing methods [4]. As such, this is an active research topic in the field of bioinformatics.

At present, ML is applied to specific data in order to explain and answer a specific biological query in the biological knowledge domain. One of the challenges of future integrative model-based ML is the ability to combine different biological resources to enhance our understanding of multiple biological questions. Once the full potential of the available data is achieved, they can be used in the development of gene-based diagnostic tests, drug discovery studies and in the development of therapeutic strategies for improving public health.

To sum up, since biological systems are quite complex and they have an interconnected nature, a single model that is trained on a single dataset can only benefit from a small portion of the entire biomedical knowledge. For this reason, in order to get the complete picture of molecular biology and medicine, the integration of diverse biological resources and multi-omics data is crucial. In the field of gene expression data analysis, there are still many challenges that the community needs to solve, such as the integration of gene expression datasets that are generated by different research groups for the same phenotype (which will help to overcome the batch effect), and an additional obstacle is the integration of non-similar data, wherein each dataset tackles a specific disease.

## Figures and Tables

**Figure 1 entropy-23-00002-f001:**
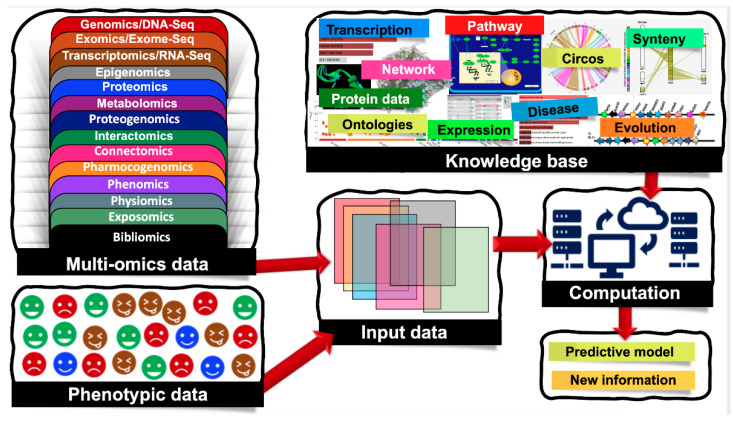
Machine learning (ML) applications that combine multi-omics and phenotypic data. Multi-omics data are classified into the following groups: genomics/DNA-Seq—the study of the genetic material for an organism, it assesses DNA sequence and structural variations including single-nucleotide polymorphisms (SNPs), insertions and deletions, copy number variations (CNVs), and inversions; epigenomics—the measurement of DNA methylation, histone modifications (methylation, acetylation, phosphorylation, DP-ribosylation, and ubiquitination), and noncoding RNAs (microRNAs, long noncoding RNAs, small interfering RNAs); transcriptomics/RNA-Seq—the study of the transcriptome of an organism; exomics/exome-seq—the study of the exome of an organism (coding regions); proteomics—the study of the total proteins within an organism; metabolomics—the study of the total metabolites; proteogenomics—combined study of genomics and proteomics; interactomics—interactions between nucleotides, proteins and metabolites; connectomics—study of the connections, neural pathways in the brain; pharmocogenomics—the application of genomics to pharmacology; phenomics—observable phenotypes; physiomics—functional behavior of an organism; exposomics—study of an organism’s environment and bibliomics (the literature concerning a topic).

**Figure 2 entropy-23-00002-f002:**
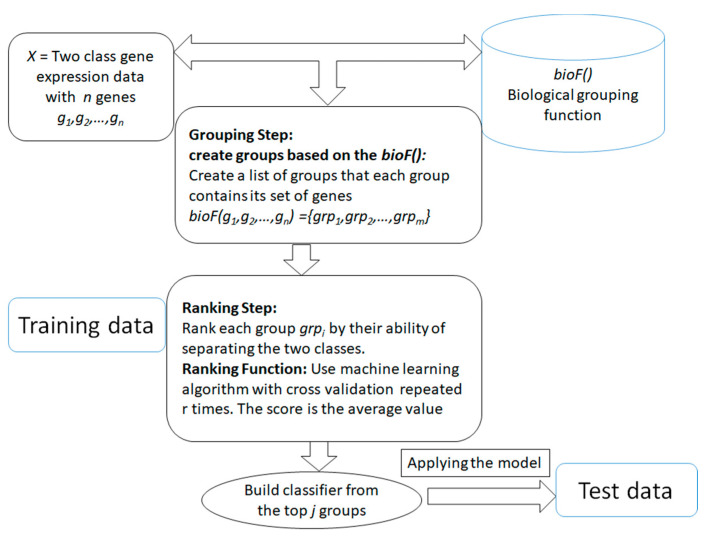
The generic framework of the algorithm that is based on biological integration for grouping, ranking and classification.

**Table 1 entropy-23-00002-t001:** Summary table of the presented methodologies that integrate biological knowledge. While “A” refers to the advantages, “D” refers to the disadvantages of the methods.

Tool Name	Incorporated Biological Knowledge	Methodology	Advantage/Disadvantage	Ref
N/A	GO	Rank the genes uses information gain (IF) incorporated with Gene Ontology GO terms	**A:** The novelty of this work is to evaluate genes based on not only their individual discriminative powers but also the powers of GO terms that annotate them.	[21]
N/A	GO	χ2, ReliefF, or IG	**A:** Including biological knowledge in the gene selection process improves results.	[22]
N/A	Combines KEGG and GO terms	Utilizes graphical causal modeling IG as an initial filter search for GO and KEGG annotations’ frequent items	**A:** Method is capable of intelligently selecting genes for learning effective causal networks. **D:** No significant improvement in accuracy.	[18]
N/A	KEGG, DisGeNET, and further genetic meta information	Gene–disease association score from DisGeNET Gene distance metrics		[23]
N/A	KEGG pathways	Uses these pathways as features for further pattern mining	**A:** Reduce the dimension of the data by transforming to KEGG feature space. **A:** Improved performance over different traditional approaches.	[24]
N/A	Gene ontology (GO)	Randomized search (CLARANS)	**A:** Reducing the dimension dramatically.	[25]
SVM-RCE	Genes related are correlated	SVM and K-means	**A:** Discover significant of clusters. **D:** Might lose important genes because they were in lower-ranked clusters.	[28,36]
SVM-RNE	GXNA for creating subnetworks from gene expression	SVM, GXNA	**A:** Reducing the dimension of the data by considering subnetworks. **D:** The subnetworks are created as a prediction of the gene expressions data.	[30]
maTE	microRNA genes targets	Random forest groups the genes that associated with microRNA	**A:** A novel approach of integrating microRNA into gene expression. **D:** The size of the groups might be large and might rank these groups highly as a result of that.	[29]
CogNet		Random forest, based on pathFindR tool	**A:** Improve the results of the pathFindR tool by ranking its groups.	[39]
miRcorrNet		Random forest based on the correlation with miRN expressions	**A:** Novel approach for integrating miRNE and mRNA expressions using machine learning.	[40]

**Table 2 entropy-23-00002-t002:** Example of microRNA and their targets list.

MicroRNA Group Name	Target Genes List
HSA-MIR-147A	VEGFA, ACVR1C, MCM3, NDUFA4, PSMA3, HIF3A, SLC22A3, MCM3, NDUFA4, PSMA3, HIF3A, VEGFA, ACVR1C, MCM3, NDUFA4, PSMA3, HIF3A, SLC22A3
HSA-MIR-18B-5P	ESR1, MDM2, CTGF, TNRC6B, HIF1A, SMAD2, FOXN1, IGF1, IGF1, CTGF, HIF1A, SMAD2, FOXN1, ESR1, MDM2, CTGF, TNRC6B, HIF1A, SMAD2, FOXN1, IGF1, IGF1
HSA-MIR-19B-3P	BACE1, PTEN, PTEN, PTEN, ATXN1, HIPK3, ARID4B, MYLIP, ESR1, KAT2B, SOCS1, BCL2L11, BCL2L11, TGFBR2, TGFBR2, BMPR2, BMPR2, TLR2, PPP2R5E, PPP2R5E, CYP19A1, GCM1, HIPK1, SMAD4, MYCN, MXD1, BCL3, DNMT1, TNFAIP3, PKNOX1, MTUS1, PITX1, PTEN, PTEN, PTEN, ATXN1, ESR1, NCOA3, KAT2B, SOCS1, TGFBR2, BMPR2, CUL5, TLR2, HIPK1, MXD1, BCL3, TNFAIP3, MTUS1, PITX1, BACE1, PTEN, PTEN, PTEN, PTEN, ATXN1, HIPK3, ARID4B, MYLIP, ESR1, NCOA3, KAT2B, SOCS1, BCL2L11, BCL2L11, TGFBR2, TGFBR2, BMPR2, BMPR2, CUL5, TLR2, PPP2R5E, PPP2R5E, CYP19A1, GCM1, HIPK1, SMAD4, MYCN, MXD1, BCL3, DNMT1, TNFAIP3, PKNOX1, MTUS1, PITX1
HSA-MIR-210-5P	CFB

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
