# Peer review of "Application of Biological Domain Knowledge Based Feature Selection on Gene Expression Data"

_entropy, 2020, doi:10.3390/e23010002_

Round 1
Reviewer 1 Report
Overall, the manuscript entitled “Application of Biological Domain Knowledge Based Feature Selection on Gene Expression Data” is a well documented review on machine learning methods applied to gene expression analysis.
Specific comments:
- Tables: Table 1 contains replicates of "Target Genes Lists" in almost every row. Please, remove replicates for a better understanding.
Author Response
Comments and Suggestions for Authors
Overall, the manuscript entitled “Application of Biological Domain Knowledge Based Feature Selection on Gene Expression Data” is a well documented review on machine learning methods applied to gene expression analysis.
>>Authors: We thank reviewer #1 for valuable suggestions and comments that helped us in enhancing the current review. We have corrected matching “word-for-word quotations” and issues related matching sentences.
Specific comments:
- Tables: Table 1 contains replicates of "Target Genes Lists" in almost every row. Please, remove replicates for a better understanding.
>>Authors: We have removed this replication.
Reviewer 2 Report
Authors have provided a review of methods of feature selection during gene expression data analysis.
- The abstract is too long and needs to be more succinct.
- Authors mention the term 'biological knowledge' a lot of times, however, they need to be more specific as it is a very general term and can mean a lot of things.
- The Introduction section is very long. Introduction should be reduced to not more than two pages.
- Authors need to provide proper classification of each of the methodologies before describing each of them in further detail.
- In section 2.2, authors mention how traditional gene selection methodologies have drawbacks, but do not elaborate on this point. Authors need to define the drawbacks before discussing the methodologies that cater to those drawbacks.
- Table 2 is first mentioned on page 7 and is them added on pages 11/12. Figures or tables should be added at the point of their first mention in the manuscript. If the table is encompassing multiple sections, authors can possibly split the information into two tables.
- The table 2 looks great!
- The last section should just labelled Conclusions. A review should cover discussion of the work during the rest of the manuscript. There is not need for a final section of discussion.
Author Response
Comments and Suggestions for Authors
Authors have provided a review of methods of feature selection during gene expression data analysis.
>>Authors: We thank reviewer #2 for valuable suggestions and comments that helped us in enhancing the current review. We have corrected matching “word-for-word quotations” and issues related matching sentences.
The abstract is too long and needs to be more succinct.
>>Authors: We have shortened it.
Authors mention the term 'biological knowledge' a lot of times, however, they need to be more specific as it is a very general term and can mean a lot of things.
>>Authors: Yes, you are right. But we do not want to be specific to any source of knowledge because in the future more discovery will be made and these discoveries will be added to the term “biological knowledge”.
The Introduction section is very long. Introduction should be reduced to not more than two pages.
>>Authors: The introduction in this study is long since we are writing a survey paper that we need to introduce several terms. We might split it to two sections, but then it will be just artificial splitting.
Authors need to provide proper classification of each of the methodologies before describing each of them in further detail.
>>Authors: We have added a paragraph to the beginning of Section 2 and 3.1. Section 3.2 already has an explanation before we explain each method in detail. We have also done that through the introduction section.
In section 2.2, authors mention how traditional gene selection methodologies have drawbacks, but do not elaborate on this point. Authors need to define the drawbacks before discussing the methodologies that cater to those drawbacks.
>>Authors: Now we have explained it in more detail in section “2.1. Traditional Gene Selection”
Table 2 is first mentioned on page 7 and is them added on pages 11/12. Figures or tables should be added at the point of their first mention in the manuscript. If the table is encompassing multiple sections, authors can possibly split the information into two tables.
>>Authors: Yes that's right- but in our case it is not critical and it is just a kind of summary to the reader.
The table 2 looks great!
>>Authors: Thank you.
The last section should just labelled Conclusions. A review should cover discussion of the work during the rest of the manuscript. There is not need for a final section of discussion.
>>Authors: Corrected.
Reviewer 3 Report
The authors have addressed my concerns, thus I am happy to recommend acceptance.
Author Response
Comments and Suggestions for Authors
The authors have addressed my concerns, thus I am happy to recommend acceptance.
>>Authors: We thank reviewer #3 for accepting our revised manuscript. We also thank reviewer #3 for previous suggestions.
This manuscript is a resubmission of an earlier submission. The following is a list of the peer review reports and author responses from that submission.
Round 1
Reviewer 1 Report
Overall, the manuscript entitled “Application of Biological Domain Knowledge Based Feature Selection on Gene Expression Data” is a well documented review on machine learning methods applied to gene expression analysis. In some parts of the text there are missing quotation marks indicating the word-for-word quotations that have been taken from the cited articles. This lack can be considered plagiarism.
Specific comments:
- Abstract: Good concision and clarity.
- Introduction: This chapter has enough information about the subject of the study.
- Materials and Methods: Theorical and practical explanations of every methods are enough.
- Discussions and Conclusions: This chapter is well written but it is recommended to discuss
which are best methods in opinion of authors.
- Tables: Table 1 is missing. Please, provide this table for review.
- Figures: In my opinion, figures are correct.
Author Response
Rev1: Overall, the manuscript entitled “Application of Biological Domain Knowledge Based Feature Selection on Gene Expression Data” is a well documented review on machine learning methods applied to gene expression analysis. In some parts of the text there are missing quotation marks indicating the word-for-word quotations that have been taken from the cited articles. This lack can be considered plagiarism.
>>Authors: We thank reviewer #1 for valuable suggestions and comments that helped us in enhancing the current review. We have corrected matching “word-for-word quotations” and issues related matching sentences.
Specific comments:
- Abstract: Good concision and clarity.
- Introduction: This chapter has enough information about the subject of the study.
- Materials and Methods: Theoretical and practical explanations of every method are enough.
- Discussions and Conclusions: This chapter is well written but it is recommended to discuss
which are the best methods in opinion of authors.
>>Authors: We thank reviewer #1.
- Tables: Table 1 is missing. Please, provide this table for review.
>>Authors: Now we added it.
- Figures: In my opinion, figures are correct.
Reviewer 2 Report
In this work, authors have reviewed methods of feature selection during gene expression data analysis. The topic of the review is very relevant. However, the review is not comprehensive.
- The direction of the review is interesting, but the authors have not provided appropriate classifications for integrative gene selection techniques/methods.
- There should be further summary (preferably in a tabular format) that provides advantages, disadvantages and the specifications of the methodologies.
- Authors have restricted themselves to only SVMs, however, this is not clear from the heading or the abstract and introduction.
- Why is there a section of materials and methods in a review?
- Grouping and ranking approaches need to have a summary and table fo their own. Please classify the techniques according to grouping or ranking approaches.
Overall, the review is not very comprehensive and needs to have a lot of more background added to it. It does not provide tool comparison and scenarios where each of the tools is appropriate. The review addresses an important topic, but needs to be more detailed.
Author Response
Rev2: In this work, authors have reviewed methods of feature selection during gene expression data analysis. The topic of the review is very relevant. However, the review is not comprehensive.
>>Authors: We thank reviewer #2 for valuable suggestions and comments that helped us in improving the current review. We have taken these points in the considerations and below provided point-to-point answers to the queries.
The direction of the review is interesting, but the authors have not provided appropriate classifications for integrative gene selection techniques/methods.
>>Authors: We have taken this point into the consideration.
There should be further summary (preferably in a tabular format) that provides advantages, disadvantages and the specifications of the methodologies.
>>Authors: We have corrected this point and provided a table (Table 2), as suggested by the reviewer.
Authors have restricted themselves to only SVMs, however, this is not clear from the heading or the abstract and introduction.
>>Authors: We would like to clarify that among the tools that are presented in section 4, only SVM-RCE and SVM-RNE use SVM, other tools use Random Forest classifier. Also, the updated version of SVM-RCE, SVM-RCE-R, uses Random Forest (RF). In general, we can't report the advantage of RF over SVM or vice versa. There is no such a comparative study.
Why is there a section of materials and methods in a review?
>>Authors: We thank the authors for this comment. We removed this title.
Grouping and ranking approaches need to have a summary and table of their own. Please classify the techniques according to grouping or ranking approaches.
>>Authors: In the revised version, we have such information in the summary table (Table 2).
Overall, the review is not very comprehensive and needs to have a lot of more background added to it. It does not provide tool comparison and scenarios where each of the tools is appropriate. The review addresses an important topic, but needs to be more detailed.
>>Authors: We are not able to provide such information since each tool has its own unique methodologies. But, we have now added a summary table indicating additional information about each methodology.
Reviewer 3 Report
In general this is a well written review, and is well on the way to publication. I did not scan for minor editorial issues (spelling, grammar, etc.), but I saw very few of those, and only a small number of awkward phrases of the sort that will likely be found and corrected with another round of proof-reading.
In terms of the technical coverage, the material represents a decent overview of the state of practical bioinformatics pursuit of omics analysis. The text is only mildly 'prospective', however.
For example, in Fig. 1, the figure caption ran through nearly the whole list of 'omics' depicted in the graphic, except for one of the items that I believe may be among the most crucial for accounting for poor predictions in prior generations of bioinformatic machine learning studies -- epigenetics/epigenomics. It is true that some of this information is conveyed indirectly through transcriptomics and other paradigms but, nonetheless, I believe that epigenetics may be crucial to fully accounting for phenomics. I believe that accounting for such molecular deviations is crucial for making tangible improvements in biomarker analysis.
Another deficiency is the apparent de-emphasis on another major point of phenotypic differentiation -- post-translational modifications of both the physiologically reasonable and the pathologically anomalous forms. These are a bit outside of the standard 'omics' realm, but methods for bioinformatically incorporating PTM effects are emerging from the gradual improvement of sequence motifs or, less directly, from compensatory expression patterns that emerge when an organism seeks to correct for aberrant biochemistry arising from anomalous structural modifications.
It is up to the authors to define the scope that they wish to address in a paper but, in practice, I doubt that informatics methods will truly be able to tackle many key biomedical challenges that they're attempting without more effective accounting for such non-canonical nuances.
Author Response
Rev3: In general this is a well written review, and is well on the way to publication. I did not scan for minor editorial issues (spelling, grammar, etc.), but I saw very few of those, and only a small number of awkward phrases of the sort that will likely be found and corrected with another round of proof-reading.
>>Authors: We thank reviewer #3 for valuable suggestions and comments that helped us in improving the current review. We have taken these points into consideration and below we have provided point-to-point answers to the queries.
In terms of the technical coverage, the material represents a decent overview of the state of practical bioinformatics pursuit of omics analysis. The text is only mildly 'prospective', however.
>>Authors: We have revised and re-edit the manuscript.
For example, in Fig. 1, the figure caption ran through nearly the whole list of 'omics' depicted in the graphic, except for one of the items that I believe may be among the most crucial for accounting for poor predictions in prior generations of bioinformatic machine learning studies -- epigenetics/epigenomics. It is true that some of this information is conveyed indirectly through transcriptomics and other paradigms but, nonetheless, I believe that epigenetics may be crucial to fully accounting for phenomics. I believe that accounting for such molecular deviations is crucial for making tangible improvements in biomarker analysis.
>>Authors: We thank the reviewer for this constructive comment. In the revised version, we have added a paragraph to the introduction section, which discusses the impact of epigenomics and how the complex cascades between genome, epigenome and transcriptome affect human diseases.
Another deficiency is the apparent de-emphasis on another major point of phenotypic differentiation -- post-translational modifications of both the physiologically reasonable and the pathologically anomalous forms. These are a bit outside of the standard 'omics' realm, but methods for bioinformatically incorporating PTM effects are emerging from the gradual improvement of sequence motifs or, less directly, from compensatory expression patterns that emerge when an organism seeks to correct for aberrant biochemistry arising from anomalous structural modifications.
It is up to the authors to define the scope that they wish to address in a paper but, in practice, I doubt that informatics methods will truly be able to tackle many key biomedical challenges that they're attempting without more effective accounting for such non-canonical nuances.
>>Authors: We thank the reviewer for this constructive comment. In the revised version, we have added a paragraph to the introduction section, which discusses the impact of post-translational modifications (PTM).